# A systematic approach to identify recycling endocytic cargo depending on the GARP complex

Sebastian Eising[1,2], Lisa Thiele[1,2], Florian Fröhlich[2]*

[1]Department of Biology/Chemistry, Molecular Membrane Biology Group, University of Osnabrück, Osnabrück, Germany; [2]Center of Cellular Nanoanalytics, University of Osnabrück, Osnabrück, Germany

**Abstract** Proteins and lipids of the plasma membrane underlie constant remodeling via a combination of the secretory- and the endocytic pathway. In the yeast endocytic pathway, cargo is sorted for recycling to the plasma membrane or degradation in vacuoles. Previously we have shown a role for the GARP complex in sphingolipid sorting and homeostasis (Fröhlich et al. 2015). However, the majority of cargo sorted in a GARP dependent process remain largely unknown. Here we use auxin induced degradation of GARP combined with mass spectrometry based vacuolar proteomics and lipidomics to show that recycling of two specific groups of proteins, the amino-phospholipid flippases and cell wall synthesis proteins depends on a functional GARP complex. Our results suggest that mis-sorting of flippases and remodeling of the lipid composition are the first occurring defects in GARP mutants. Our assay can be adapted to systematically map cargo of the entire endocytic pathway.
DOI: https://doi.org/10.7554/eLife.42837.001

**\*For correspondence:**
florian.froehlich@biologie.uni-osnabrueck.de

**Competing interests:** The authors declare that no competing interests exist.

## Introduction

The plasma membrane forms the boundary of cells that mediates all communication and transport in and out of the cell. To maintain these complex functions the composition of the plasma membrane is highly regulated and needs constant remodeling. Plasma membrane proteins and lipids are taken up by endocytosis and are delivered to early endosomes. Endosomes are the main sorting station in the endosomal pathway. These compartments are necessary for sorting, recycling and degradation of cargo molecules (*Maxfield and McGraw, 2004*).

Proteins destined for degradation are sorted by the ESCRT complexes (*Henne et al., 2011*) into multivesicular bodies (MVBs) which are finally fused with the vacuole. The process of endosomal maturation requires switching from the Rab5 to the Rab7 GTPase as well as a change in the phosphoinositide composition (*Cabrera and Ungermann, 2010*; *Huotari and Helenius, 2011*). For the final fusion of endosomes with the lysosome/vacuole the so called HOPS (homotypic fusion and vacuole protein sorting) tethering complex and the RAB family GTPase Ypt7 are required.

An alternative pathway for endosomal cargo is the recycling pathway to the plasma membrane. In mammalian cells, specialized recycling endosomes recycle cargo back to the plasma membrane (*Maxfield and McGraw, 2004*). In yeast cells, recycling of endocytic cargo to the plasma membrane requires the Golgi apparatus. Depending on different sorting complexes such as the retromer (*Seaman et al., 1998*) or the Snx4/41/42 complex (*Ma et al., 2017*) proteins can be targeted to the Golgi. Recent discoveries suggest that the trans-Golgi network in yeast also serves as the early/recycling endosome. Additionally, yeast harbors a separate late/prevacuolar endosome (*Day et al., 2018*). In this model, endocytic cargo is delivered directly to the trans-Golgi network (TGN) and is further sorted for recycling or transport to the prevacuolar compartment.

For the tethering of retrograde endosomal transport carriers at the Golgi the GARP (Golgi associated retrograde protein trafficking) complex is required. GARP is a hetero-tetrameric complex consisting of the four subunits Vps51, Vps52, Vps53 and Vps54 and belongs to the family of CATCHR (complexes associated with tethering containing helical rods) complexes (*Chou et al., 2016*; *Conibear and Stevens, 2000*; *Siniossoglou and Pelham, 2002*; *Vasan et al., 2010*). Deletion of the GARP complex has been linked to multiple cellular dysfunctions. The first discovered and canonical pathway is the sorting of the carboxy peptidase Y (CPY) receptor Vps10 (*Conibear and Stevens, 2000*), hence the name Vps of all subunits. However, deletion of the GARP complex has also been linked to defects in autophagy and mitochondrial tubulation (*Reggiori and Klionsky, 2006*), defects in the actin cytoskeleton (*Fiedler et al., 2002*), cell wall integrity (*Conde et al., 2003*), vacuole integrity (*Conibear and Stevens, 2000*) and several more (for a complete overview see *Bonifacino and Hierro, 2011*).

We have previously identified an important role for the GARP complex in lipid homeostasis (*Fröhlich et al., 2015*). Deletion of either subunit of the GARP complex results in the massive accumulation of sphingolipid intermediates, the long chain bases (LCBs) in cells. Interestingly all observed defects in GARP knockout mutants, including vacuolar fragmentation can be rescued by chemical depletion of sphingolipids. This suggests that sphingolipid accumulation is the causing problem in cells but the molecular mechanism for this remains largely elusive.

Here, we have developed a system that combines auxin induced degradation of the GARP complex with mass spectrometry based vacuolar proteomics and lipidomics to systematically identify cargo of the GARP dependent endosomal sorting pathway. We show that plasma membrane proteins of two different functional groups, amino-phospholipid flippases and cell wall biosynthesis proteins, are the first to be mis-sorted after chemical depletion of the GARP complex. We also analyze the cellular and vacuolar lipid composition to shed some light on the important functions of the GARP complex in cells.

## Results

### Acute GARP inactivation by auxin-mediated degradation

Deletion of any subunit of the hetero-tetrameric GARP complex results in a plethora of cellular phenotypes ranging from endosomal sorting defects, to mitochondrial dysfunction, to problems with the CVT pathway and cellular sphingolipid accumulation (*Bonifacino and Hierro, 2011*; *Fröhlich et al., 2015*). However, with its canonical role in retrograde endosome to Golgi trafficking, the first occurring changes causing the aforementioned defects remain unknown. To study the causing defects occurring in cells after deletion of the GARP complex we made use of the auxin induced degron (AID) system. We therefore tagged the Vps53 subunit of the GARP complex C-terminally with an AID tag followed by a 6 HA tag. To enable auxin-dependent recognition of Vps53-AID-6HA by an ubiquitin ligase, we expressed the Oryza sativa F-box transport inhibitor response-one auxin receptor protein (OsTir1) in these cells. In immunoblot experiments we could detect similar protein levels for Vps53-AID-6HA compared to Vps53 only carrying a C-terminal 6 HA tag (*Figure 1a*). Addition of the auxin analogue 3-idoleacetic acid (IAA) resulted in the rapid degradation of the AID tagged Vps53 (*Figure 1a*). Quantification of several replicates revealed that 90% of the protein as degraded after 10 min. After 60 min 99% of the protein are degraded which we consider complete degradation (*Figure 1a,b*).

Mutations in the GARP complex accumulate large amounts of the sphingolipid intermediate dihydrosphingosine and show strong growth defects. This can be reversed by addition of the serine palmitoyltransferase inhibitor myriocin to the growth medium (*Fröhlich et al., 2015*). To test if auxin induced degradation of the GARP subunit Vps53 resembled the phenotype of the knockout we spotted cells on plates containing myriocin, IAA or a combination of both. On control plates WT cells, cells expressing only OsTir, cells expressing only the AID-tagged Vps53 and cells expressing both, the ubiquitin ligase and the AID tag on Vps53 showed normal growth, whereas *vps53Δ* showed a growth defect (*Figure 1c*, upper left panel). On plates containing IAA the Vps53-AID OsTir strain showed a slight growth defect (*Figure 1c*, upper right panel). As expected, only the *vps53Δ*

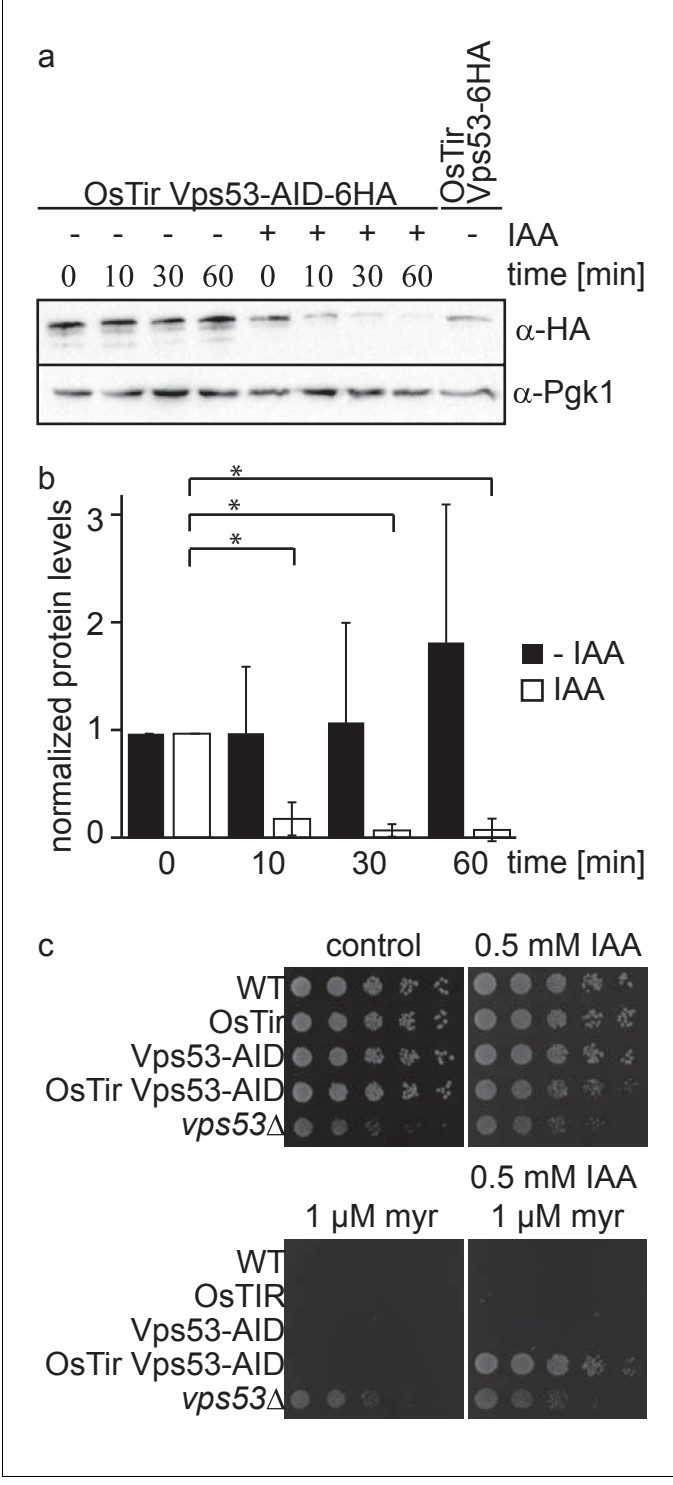

**Figure 1.** Vps53 can be depleted using the auxin induced degron system. (**a**) AID tagged Vps53 is rapidly degraded after addition of IAA. OsTir Vps53-AID-6HA cells were treated with IAA or ethanol (control) for the indicated times. Cells were lysed and equal amounts of proteins were loaded and analyzed by western blotting using antibodies against the HA tag or Pgk1 as a loading control. An OsTir Vps53-6HA strain was used as a control to exclude any effect of the AID tag on protein abundance. (**b**) Quantification of a (n = 7 experiments). Vps53-AID-HA band intensities were quantified and normalized to Pgk1 signals. Vps53-AID-6HA levels start to significantly decrease after 10 min of IAA induction (p=0.000009824). (**c**) Auxin induced degradation of Vps53 pheno-copies a *VPS53* deletion. Wild-type cells, cells expressing OsTir, cells harboring the Vps53-AID-6HA tag, cells expressing

*Figure 1 continued on next page*

*Figure 1 continued*

OsTir and Vps53-AID-6HA and *vps53Δ* cells were serial diluted on control plates, plates containing 500 µM IAA, plates containing 1 µM myriocin and plates containing 500 µM IAA and 1 µM myriocin.
DOI: https://doi.org/10.7554/eLife.42837.002

strain grew on plates containing myriocin (*Figure 1c*, lower left panel). On plates containing a combination of IAA and myriocin the Vps53-AID OsTir strain started to grow again, showing that IAA addition to this strain results in a functional knockout (*Figure 1c*, lower right panel).

## GARP inactivation results in vacuolar fragmentation

With a chemically inducible knockout of the GARP complex we wanted to test the impact of the loss of a functional GARP complex on the cell and its organelles. GARP knockouts cells show very strong vacuolar fragmentation phenotypes. One hypothesis is that loss of GARP function results in a decrease in recycling from endosomes via the Golgi to the plasma membrane and therefore accumulation of cargo at the vacuole. One potential cargo are LCBs resulting from the breakdown of complex sphingolipids which are speculated to cause the vacuolar defects. To test the effect of acute GARP inactivation on the vacuole we tagged the vacuolar membrane protein Vph1 with a GFP tag in cells expressing Vps53-AID-HA and OsTir. In a control strain harbouring Vps53-AID-6HA without OsTir we labelled Vph1 with a mCherry tag. To determine the effect of Vps53 degradation on the vacuole we mixed the two strains of the same mating type, added IAA to the cells and monitored the vacuolar morphology over time (*Figure 2a*). Yeast cells usually carry one to three round vacuoles, as we observed for more than 70% of the cells in both strains under conditions without IAA (*Figure 2b*). Over time, the addition of IAA caused an increase of cells harbouring more than three vacuoles as early as 30 mins after addition of IAA only in the strain carrying Vps53-AID-6HA and OsTir. After 90 mins of treatment this number increased to more than 65%. In contrast, control cells showed no change in the vacuolar morphology over time showing that the inactivation of GARP function rapidly results in changed vacuolar morphology (*Figure 2b*). However, this phenotype is not as strong as a *VPS53* deletion suggesting that the accumulation of cargo enhances the phenotype (*Fröhlich et al., 2015*).

Taken together, we hypothesized that we can identify protein and lipid cargo that is transported in a GARP dependent manner in or on purified vacuoles from cells where we chemically induced GARP depletion. To test if we can purify vacuoles from yeast cells with chemically depleted Vps53 we tagged Vph1 with either a GFP tag or a mCherry tag in the Vps53-AID strain. We induced Vps53 depletion in the strain harbouring Vph1-GFP with IAA for 90 min. We next mixed both strains and purified vacuoles according to established protocols (*Cabrera and Ungermann, 2008*). Purified vacuoles were analyzed by live cell imaging to determine the number of purified vacuoles as well as their average diameter (*Figure 2c and d*). We observed a mild increase in the number of purified vacuoles from IAA treated cells together with a concomitant decrease in the average vacuolar diameter (*Figure 2d*), suggesting that vacuoles can still be purified from GARP depleted cells. It is possible that vacuoles that are further fragmented are lost during purification. Together, our data show that vacuoles from yeast cells with a chemically depleted GARP complex can be purified in similar numbers as to untreated control cells. This suggests that early changes in the vacuolar proteome and lipidome in GARP depleted cells can be determined systematically.

## Enriched vacuoles can be characterized by MS based proteomics

To study the cargo that is transported to the vacuole in GARP depleted strains we first established the methods for purification of the vacuoles and their analysis using SILAC (*Ong and Mann, 2007*) labeling followed by MS based proteomics. Yeast vacuoles have been characterized by mass spectrometry based proteomics before (*Wiederhold et al., 2009*). In this study 77 proteins were identified that were annotated as vacuolar proteins, equalling 42% of all annotated vacuolar proteins. In total 13% of all identified proteins belonged to the vacuolar fraction. To determine the purity of vacuoles in our hands, we purified vacuoles from lysine 0 labelled cells and mixed them with total cell lysates from lysine 8, 'heavy' labelled cells (*Figure 3a*). Our analysis resulted in a total of 1599 proteins yielding a SILAC ratio (*Figure 3—source data 1*). Of these proteins 135 were identified

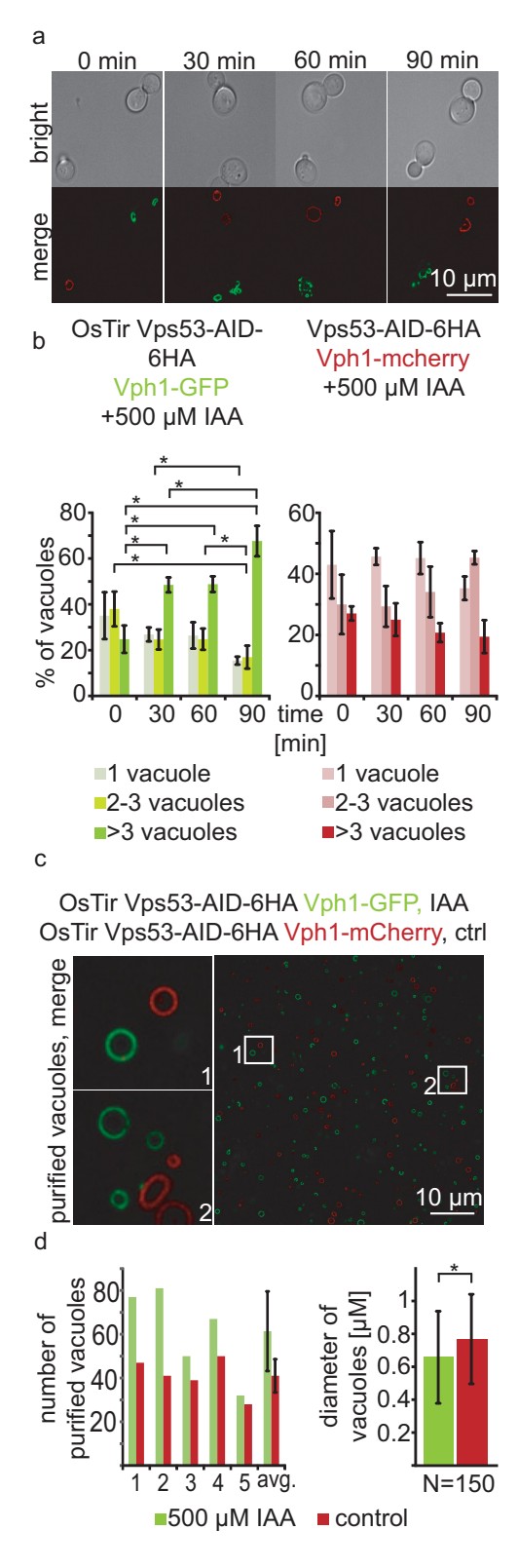

**Figure 2.** Acute depletion of Vps53 leads to vacuolar fragmentation. (a) Yeast cells expressing Vps53-AID-6HA together with the OsTir ligase and Vph1-GFP (green vacuoles) and cells expressing only Vps53-AID-6HA and Vph1-mCherry were mixed and treated with 500 μM IAA for 0, 30, 60 and 90 mins. Brightfield images (upper panels) and merged images are shown (lower panels). Only cells expressing OsTir and Vps53-AID-6HA show

*Figure 2 continued on next page*

*Figure 2 continued*

fragmented vacuoles. Scale bar = 10 µM (b) Quantification of a. Vacuole fragmentation from three different experiments (error bars show standard deviation). Shown is the percentage of cells with one vacuole, 2–3 vacuoles or more than three vacuoles (green = a strain harboring OsTir Vps53-AID-6HA; red = a strain harboring only Vps53-AID-6HA). The amount of cells with more than three (=fragmented) vacuoles increases over time in the functional AID-strain (n = 30–200 cells per setting) (c) Vacuoles can be purified after auxin induced degradation of Vps53. Fluorescent microscopy of purified vacuoles from IAA or mock treated Vps53-AID-6HA strains. Cells with the functional AID-strain and GFP-marked vacuoles were incubated with IAA. Cells with the functional AID-strain and mCherry-marked vacuoles were mock treated with EtOH as control. Vacuoles were isolated from the two strains that were mixed prior to lysis. (d) Quantification of the amount (left) and the diameter (right) of purified vacuoles. In average from five different samples the amount of vacuoles from IAA induced cells is slightly higher than control vacuoles. Purified vacuoles from mock treated and IAA treated cells show no significant difference in vacuolar diameter.

DOI: https://doi.org/10.7554/eLife.42837.003

that are annotated as vacuolar proteins. To carefully analyze our data and identify contaminants we plotted the SILAC ratios of all identified proteins against their total intensities. As expected we observed a strong bias towards proteins with low SILAC ratios reflecting proteins that are enriched in the vacuole preparations. We next binned the proteins according to their logarithmic SILAC ratios and analyzed the GO (gene ontology) terms of the different ratio bins. This analysis revealed that proteins in the two bins with the lowest SILAC ratios ($log_2$ ratio −5 to −6 and −4 to −5) were significantly enriched in proteins with the GO term 'vacuole' (p<$4.86^{-28}$ and p<$4.26^{-14}$, *Figure 3b and c*). The next two higher ratio bins ($log_2$ ratio −4 to −5 and −2 to −4) also yielded proteins annotated as vacuolar but with lower significance values. Instead these bins were highly enriched for ER and lipid droplet localized proteins (*Figure 3c*). Together, these data show that we can enrich vacuolar proteins in our preparations. However, together with vacuoles we also enriched proteins from the ER and lipid droplets. This supports the idea that yeast vacuoles form extensive membrane contact sites with both organelles (*Bouchez et al., 2015*; *Pan et al., 2000*; *van Zutphen et al., 2014*).

## Acute GARP inactivation results in the accumulation of amino-phospholipid flippases and cell wall synthesis proteins at the vacuole

The canonical function for the GARP complex is the tethering of retrograde endosome to Golgi transport carriers (*Conibear and Stevens, 2000*). Recent discoveries suggest that the TGN can act as the early/recycling endosome in yeast, thus GARP can potentially tether endocytic vesicles originating from the plasma membrane. Independent of the pathway, we hypothesized that transport carriers that are not tethered at the Golgi in a GARP deficient strain will eventually arrive at the yeast vacuole. We should thus be able to identify the cargo by purifying vacuoles from a GARP depleted strain and compare their protein content to a control strain. Since vacuoles are completely fragmented in a GARP knockout strain we purified vacuoles from an OsTir Vps53-AID-6HA strain labelled with heavy lysine and treated with IAA and compared them to vacuoles isolated from a mock treated, light lysine labelled OsTir Vps53-AID-6HA strain. To also control the overall protein levels in the cells we quantified the entire proteome of the cell prior to vacuole purification (for the experimental setup see *Figure 4a*). To ensure that we identify mis-targeted proteins as early as possible we analyzed the vacuolar and cellular proteome after 0, 30, 60 and 90 min of IAA treatment. The most pronounced phenotypes were observed after 90 min of IAA treatment. In the purified vacuole sample we identified a total of 1515 proteins with yielding a SILAC ratio (t = 90 min; *Figure 4—source data 1*). From these proteins 78 showed a significantly increased heavy/light SILAC ratio (according to significance A [*Cox and Mann, 2008*], p<0.05; *Figure 4b*). In line with our hypothesis that plasma membrane proteins, following a GARP dependent transport route cannot be recycled, the most enriched group of proteins are annotated as plasma membrane proteins (GO:0005886; p-value<$4.74^{-6}$, 2.53 fold enrichment). Amongst these proteins are especially the plasma membrane localized amino-phospholipid flippases Dnf1 and Dnf2 as well as their adaptor protein Lem3 (*Hachiro et al., 2013*). Another group of enriched proteins, according to GO term analysis, belonged to fungal cell wall proteins (GO:0009277; P value > $2.71^{-5}$, 5.9 fold enrichment). Interestingly, both phospholipid flipping and cell wall maintenance have been previously linked to a

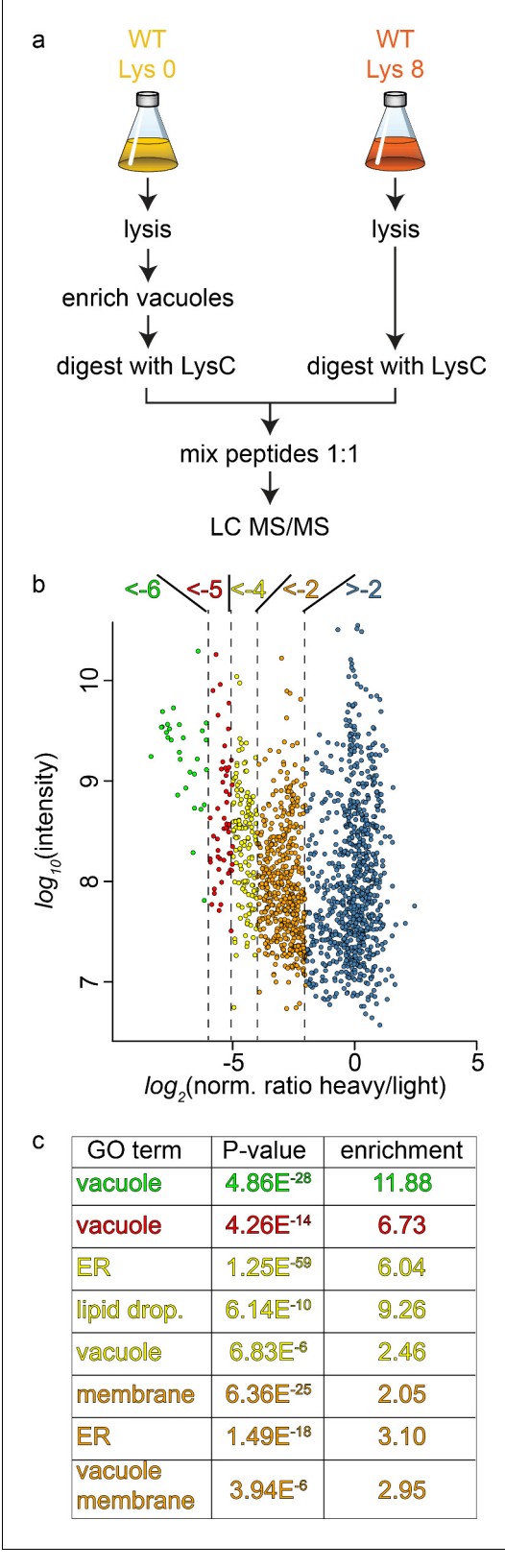

**Figure 3.** Proteomic analysis of enriched yeast vacuoles. (a) Experimental setup to determine vacuolar enrichment. (b) Proteomic analysis of purified vacuoles mixed whole cell lysates. Proteins are color coded according to ratio bins (<-6, green; −6->5, red; −5->4, yellow; −4->2, orange; −2- > 5, blue). Protein intensities are plotted
*Figure 3 continued on next page*

*Figure 3 continued*
against heavy/light SILAC ratios. (c) Enriched GO terms in the ratio bins from b. Go terms were calculated according to the Gene Ontology enRIchment anaLysis and visuaLizAtion tool, GOrilla.
DOI: https://doi.org/10.7554/eLife.42837.004
The following source data is available for figure 3:
**Source data 1.** List of all proteins identified including SILAC ratios and intensities.
DOI: https://doi.org/10.7554/eLife.42837.005

functional GARP complex (*Conde et al., 2003*; *Takagi et al., 2012*). The complete list of proteins identified including the significant outliers is provided in *Figure 4—source data 1*.

Motivated by our initial results we first wanted to make sure that the OsTir ubiquitin ligase itself nor the addition of IAA had any specific effect on the proteome of the cell in general and the vacuolar proteome specifically. Therefore, we analyzed the proteomes of cell lysates and isolated vacuoles from yeast cells harboring the OsTir ubiquitin ligase treated with IAA or mock treated. We did not observe changes in any of the proteins significantly enriched in the cells were we degraded the GARP complex, except for Pdr12 which was enriched in the cell lysate of IAA treated cells compared to controls (*Figure 4—source data 2*). We also did not observe changes in the previously identified target proteins when we compared IAA treated cells with or without the expression of the OsTir ubiquitin ligase (*Figure 4—source data 2*). Together these control experiments suggest that the proteins we find enriched at vacuoles are depending on transport by the GARP tethering complex.

To further validate our results, we compared OsTir Vps53-AID-6HA cells treated with IAA for 90 mins with mock treated OsTir Vps53-AID-6HA cells again. This time we switched SILAC labeling to exclude any effects on protein trafficking from heavy or light labelled lysine. This also allows us to plot the heavy to light SILAC ratios of the two experiments against each other. The two experiments are strongly anti correlated, as expected by switching SILAC labels. Importantly, many of the previously detected proteins mis-targeted to the vacuole still remain strong outliers (*Figure 4c*).

Finally, we tested if the deletion of the vacuolar proteinase A gene, *PEP4*, had an impact on the observed proteins. If proteins reach the vacuole and are internalized, they are broken down by vacuolar proteases. Thus, it is difficult to pick up peptides of the proteins by MS based proteomics, since only peptides resulting from LysC digestion are searched for in the experiments. A deletion of the major vacuolar protease could therefore improve the identification rate of proteins that are enriched in vacuoles. Overall, results from this experiment improved the identification rate of proteins in the vacuolar samples. We again saw differences in the amino phospholipid-flippase proteins Dnf1, Dnf2 and Lem3 and in cell wall proteins. In comparison to the previous results, in these experiments we observed a vacuolar enrichment of the SNARE Tlg2 and the protein Vps45 which are known interactors of the GARP complex (*Dulubova et al., 2002*) (*Figure 4d*). However, the number of peptides of the two mentioned proteins compared to for example the flippases are relatively low (*Figure 4—source data 3*).

Together our data show that the amino-phospholipid flippase proteins of the plasma membrane are specifically re-routed to the vacuole in cells were we induce the depletion of the GARP complex. For a protein that is shifted from a recycling pathway to a degradation pathway, we expected that the overall protein levels in the cell are decreased, while the amount on or in the vacuolar fraction increased. We only observed a small decrease in total cell lysates for Dnf1 (*Figure 4e*). The protein levels for Lem3 were below the detection limit in the cell lysate. In contrast, we saw the overall protein levels of Pdr12 increasing in both, the vacuolar and the cell lysate sample. This suggests that the expression of the multi-drug transporter Pdr12 is increased by the addition of IAA (*Figure 4e*). This also leads to the conclusion that Pdr12 is probably transporting IAA out of the cell and a *pdr12Δ* strain could be more sensitive to lower levels of IAA. This has to be evaluated in the future. Interestingly, we did not observe any changes in the best described GARP dependent protein, the CPY receptor Vps10. In both the vacuolar fraction and the cell lysate the levels of Vps10 remain unchanged (*Figure 4e*). One explanation for this observation is that the levels of Vps10 at the vacuole are already high in mock treated cells. This is exactly what we observe while we did not detect any Dnf2 signal at the vacuole under these conditions.

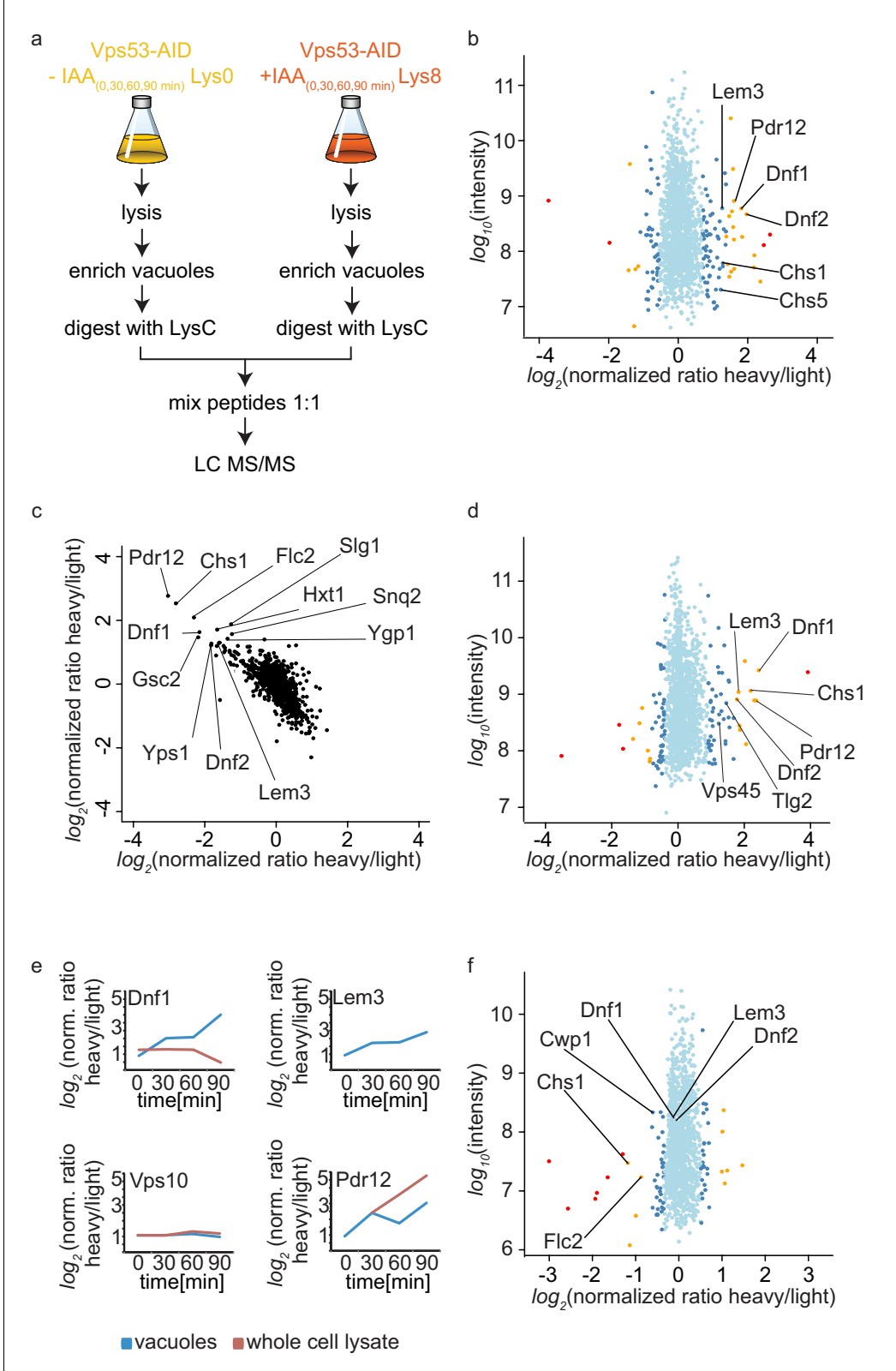

**Figure 4.** Mass spectrometry based proteomics to identify proteins mis-targeted to vacuoles in GARP depleted cells. (a) Experimental setup to determine proteins mis-sorted to the vacuole in GARP complex depleted cells. (b) Amino-phospholipid flippases are targeted to the vacuole after 90 min of IAA induced GARP depletion. Proteomic analysis of vacuoles from IAA and mock treated OsTir Vps53-AID-6HA cells is shown. Protein intensities are plotted against heavy/light SILAC ratios. Significant outliers are colored in red ($p < 1-^{11}$), orange ($p < 1-^{4}$), or steel blue ($p < 0.05$); other proteins are

*Figure 4 continued on next page*

*Figure 4 continued*

shown in light blue. (**c**) Control experiment for b). A label switching experiment of IAA induced depletion of GARP followed by proteomics analysis of the vacuoles is shown. Heavy to light ratios from an experiment where lysine 8 labelled cells were treated with IAA are plotted on the x axis vs the same experiment where the lysine 0 labelled cells were treated with IAA on the y axis. (**d**) Same experiment as in b) but in *pep4Δ* strains. (**e**) Heavy to light ratio profiles of four different proteins are shown over the time-course of IAA treatment. SILAC ratios of the vacuolar samples are shown in blue, SILAC ratios of the entire cell extract are shown in red. Note that for the flippase Dnf2 the vacuolar ratio increases while the ratio for the entire cell extract decreases. Vps10 does not show any difference in both. Pdr12 ratios increase in both samples. (**f**) The AP-2 pathway is the cargo adaptor for cell wall proteins but not for amino-phospholipid flippases. Proteomic analysis of vacuoles from IAA treated, light labelled OsTir Vps53-AID-6HA cells and heavy labelled OsTir Vps53-AID-6HA *apl1Δ* cells is shown. Protein intensities are plotted against heavy/light SILAC ratios. Significant outliers are colored in red ($p<1-^{11}$), orange ($p<1-^{4}$), or steel blue ($p<0.05$); other proteins are shown in light blue.

DOI: https://doi.org/10.7554/eLife.42837.006

The following source data and figure supplement are available for figure 4:

**Source data 1.** List of all proteins identified including SILAC ratios and intensities from the time course of auxin induced Vps53 degradation.
DOI: https://doi.org/10.7554/eLife.42837.008
**Source data 2.** List of all proteins identified including SILAC ratios and intensities from the control experiments.
DOI: https://doi.org/10.7554/eLife.42837.009
**Source data 3.** List of all proteins identified including SILAC ratios and intensities from auxin induced Vps53 degradation in a *pep4Δ* background.
DOI: https://doi.org/10.7554/eLife.42837.010
**Source data 4.** List of all proteins identified including SILAC ratios and intensities from auxin induced Vps53 degradation in an *apl1Δ* background.
DOI: https://doi.org/10.7554/eLife.42837.011
**Figure supplement 1.** Analysis of Lem3-mNeon localization depending on the AP-2 complex.
DOI: https://doi.org/10.7554/eLife.42837.007

We also hypothesized that the method we have developed to map endocytic recycling cargo could be used to identify the endocytic adaptors for the proteins that are recycled. Flippases have previously been linked to the Sla1 dependent endocytosis (*Liu et al., 2007*). However, we were not able to purify sufficient amounts of vacuoles from *sla1Δ* cells to analyse their proteomic composition. To test our hypothesis alternatively, we deleted the AP-2 adaptor complex subunit *APL1* in our functional Vps53-AID strain. Proteomic analysis of enriched vacuoles from '*light*' labelled OsTir Vps53-AID-6HA cells compared to '*heavy*' labelled OsTir Vps53-AID-6HA *apl1Δ* cells, both treated with IAA for 90 min revealed that the amino-phospholipid flippases Dnf1, Dnf2 and Lem3 were identified at the vacuole in both strains, yielding a SILAC ratio of approximately 1 (*Figure 4f* and *Figure 4— source data 4*). We confirmed or results by live cell imaging of mNeon tagged Lem3 where we could still detect Lem3 at vacuoles in IAA treated Vps53-AID-6HA cells. (*Figure 4—figure supplement 1a and b*). To exclude that the protein is targeted to the vacuole from an intracellular pool we analyzed the cellular distribution of Lem3-mNeon in OsTir Vps53-AID-6HA and OsTir Vps53-AID-6HA *apl1Δ* cells using the radial profile plot imageJ plugin. This analysis shows that the plasma membrane signal of Lem3 is depleted in cells with vacuolar Lem3 accumulation (*Figure 4—figure supplement 1c*). Only three proteins involved in cell wall maintenance that we have identified as enriched in vacuoles in GARP depleted cells (Chs1, Cwp1 and Flc2) are lower abundant at vacuoles from AP-2 deleted OsTir Vps53-AID-6HA cells (*Figure 4f*). This suggests that proteins involved in cell wall maintenance are constantly shuttled between the plasma membrane and the Golgi using AP-2 as the endocytic adaptor complex. The endocytic adaptor for the amino-phospholipid flippases remains unknown. However, this observation confirms that our method is useful to detect adaptor complexes for endocytic proteins.

## Dnf1, Dnf2 and Lem3 are targets of GARP dependent recycling

The amino phospholipid flippases Dnf1 and Dnf2 as well as the adaptor protein Lem3 are integral parts of the plasma membrane and responsible for generating phospholipid asymmetry across the plasma membrane (*Hachiro et al., 2013*; *Nakano et al., 2008*). They have been linked to endocytosis as well as sphingolipid homeostasis (*Hachiro et al., 2013*; *Roelants et al., 2010*). It has been suggested that Dnf1 and Dnf2 constantly shuttle between the plasma membrane and endosomes and this transport has been previously linked to the GARP complex (*Takagi et al., 2012*). Our MS results suggested that both, Dnf1 and Dnf2 as well as Lem3 are not recycled in GARP depleted cells and instead are re-routed to the vacuole. To confirm our MS data we tagged each of the proteins with

the mNeon green fluorescent protein (*Shaner et al., 2013*) in cells also harboring the vacuole local-ized Vph1-mCherry as well as OsTir and the AID-tagged Vps53. While we were unable to detect any signal for Dnf1-mNeon, Dnf2 and Lem3 both localized to the plasma membrane with an enrichment at either the bud or the bud neck and also some puncta (*Figure 5a,c*). Importantly, we did not observe any co-localization of the dots with Vph1-mCherry labelled vacuoles. When we compared IAA treated to mock treated cells after 30, 60 and 90 min of treatment we observed an increasing number of cells that showed Dnf1 or Lem3 signal co-localizing with the vacuole which we also quan-tified (*Figure 5a,b,c,d*). This phenotype was always observed in cells that had fragmented vacuoles. In comparison, mock treated cells showed no increase in vacuolar Lem3 or Dnf2 signal as well as no increase in vacuolar fragmentation. To control that IAA treatment itself did not affect our results we analyzed Dnf2 and Lem3 localization in IAA treated cells that were either harboring only OsTir or OsTir and an AID tagged version of Vps53. Here, we also observed Dnf2 and Lem3 localization to the vacuole only in Vps53-AID tagged cells treated with IAA, thus ruling out any side effects from the IAA treatment *Figure 5—figure supplement 1a,b*).

Interestingly, both Dnf2 and Lem3 localization to the fragmented vacuoles appear to colocalize with the Vph1-mCherry. This suggests that both proteins are not delivered to the vacuolar lumen but instead localize to the vacuolar membrane. To test this hypothesis we tagged Dnf2, Lem3, Pdr12 and as a control Itr1 C-terminally with a *pho8Δ60* in a strain lacking both *PHO8* and *PHO13*. If the proteins are delivered to the vacuolar lumen, *pho8Δ60* is cleaved and becomes active. This activ-ity can be measured in a Pho8 assay (*Yao et al., 2018*). As expected, the inositol transporter Itr1 shows an increase in Pho8 activity after addition of inositol. In contrast, neither Dnf2 nor Lem3 or Pdr12 tagged with *pho8Δ60* showed an increase in Pho8 activity suggesting that the proteins are not delivered to the vacuolar lumen (*Figure 5e*).

We also analyzed the localization of mNeon tagged Pdr12 and Vps10. As MS data suggested, the expression levels of Pdr12 are massively increased upon treatment of the cells with IAA (*Fig-ure 5—figure supplement 1d*). This confirms our hypothesis, that Pdr12 is the main transporter for IAA out of the cell. Also in line with our MS data, we did not observe any changes in the localization of Vps10 after the indicated times of IAA treatment (*Figure 5—figure supplement 1c*).

## Depletion of the GARP complex pheno-copies a Lem3 deletion

The results we obtained from both, MS based proteomics and live cell imaging suggest that Dnf1, Dnf2 and their adaptor protein Lem3 are mis-targeted to the vacuole in cells where the degradation of the GARP complex is initiated. This suggests, that the phenotypes observed in a *lem3Δ* strain should be pheno-copied in strains were the GARP complex is depleted. To test this hypothesis we first analyzed correlation coefficients of LEM3 and GARP subunits from high throughput chemical-genomics screens (*Hoepfner et al., 2014*). This analysis shows very high correlating profiles of *LEM3* and *VPS52* with other genes involved in the GARP dependent recycling pathway, such as *TLG2* and *VPS45* (*Figure 6a*).

LEM3 deleted cells are highly resistant to the cytotoxic phosphatidylcholine (PC) analog miltefo-sine (*Puts et al., 2012*). We therefore spotted WT cells, *vps53Δ* cells, *lem3Δ* cells, cells expressing OsTir and cells expressing both, OsTir and Vps53-AID on control plates, plates containing IAA, plates containing miltefosine and plates containing both, miltefosine and IAA. On control plates and IAA plates cells grew as expected. Only a *vps53Δ* strain showed a growth defect under these condi-tions, as reported previously. The addition of miltefosine resulted in a complete growth arrest in all strains, except *lem3Δ* and *vps53Δ* (*Figure 6b*). This already suggests that the *vps53Δ* strain pheno-copies a deletion of LEM3. The addition of both, IAA and miltefosine resulted in the additional growth of the Vps53-AID strain also expressing OsTir, thus confirming that the strain loses the func-tionality of Lem3 because of its transport to the vacuole. We also tested if the overexpression of Lem3 rescues the observed phenotype. The Vps53-AID strain overexpressing Lem3 from the TEF promotor also grew in the presence of IAA and miltefosine (*Figure 6b*). This suggests that the observed phenotype is also dependent on the two proteins Dnf1 and Dnf2 that form a functional complex with Lem3. Overexpression of the adaptor protein should therefore not rescue the phenotype.

We have previously shown that the deletion of the GARP subunit *VPS53* results in severe changes in the sphingolipid composition in the cell (*Fröhlich et al., 2015*). The observation that depletion of the GARP complex results in the mis-targeting of all plasma membrane amino-phospholipid

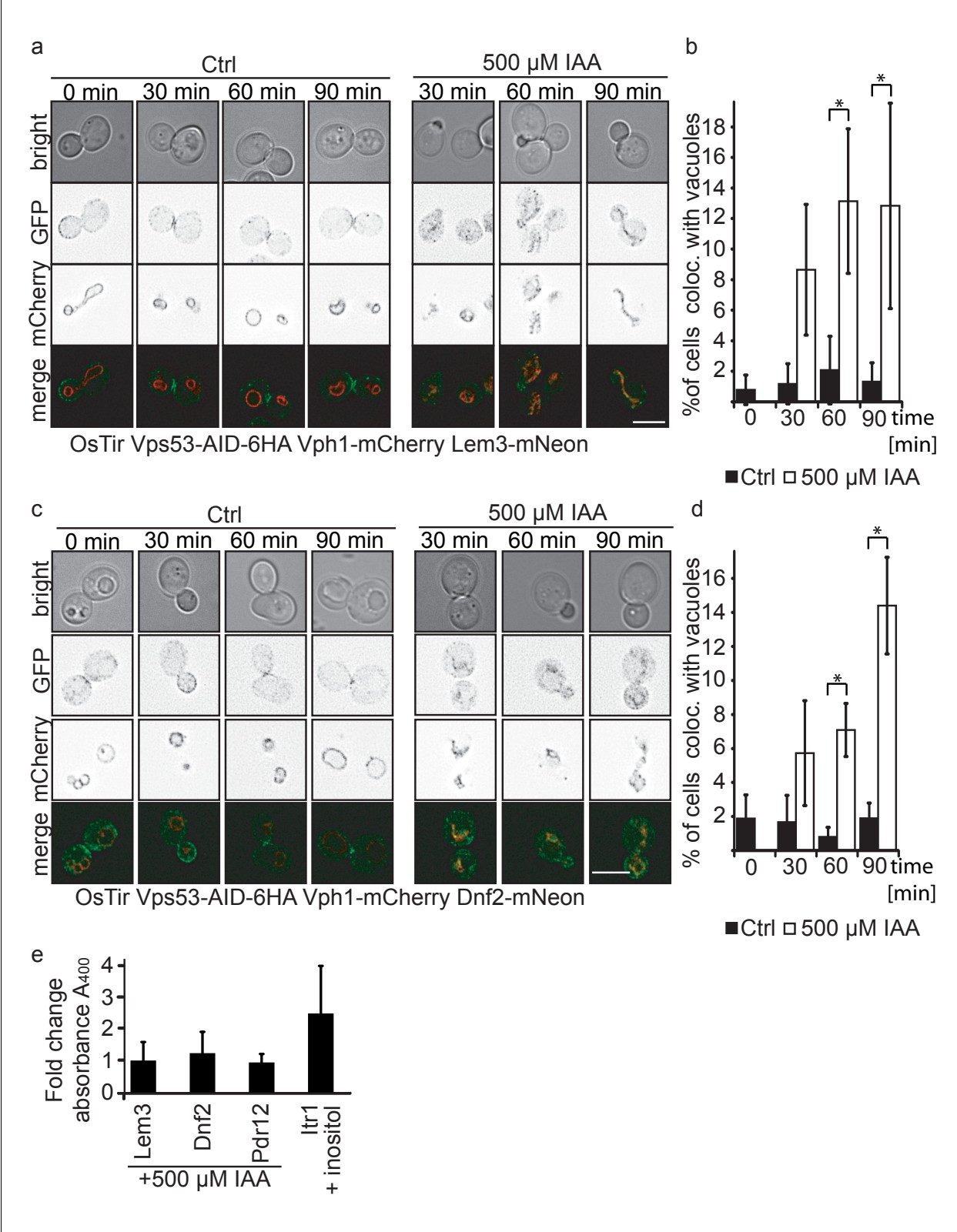

**Figure 5.** Plasma membrane localized flippases re-localize to vacuoles in GARP complex depleted cells. (**a**) Co-localization of mNeon-tagged Lem3 (second row from top) with mCherry-tagged Vph1 (third row from top) are shown in mock treated and IAA treated cells. Representative mid-sections are shown. Scale bar = 5 µM (**b**) Quantification of 4 different experiments shows co-localization of vacuoles and Lem3 in approximately 15% of cells after 90 min (error bars show standard deviation). (**c**) Co-localization of mNeon-tagged Dnf2 (second row from top) with mCherry-tagged Vph1 (third row from top) are shown in mock treated and IAA treated cells. Representative mid-sections are shown. Scale bar = 5 µM (**c**) Co-localization of mNeon-tagged Dnf2 (second row from top) with mCherry-tagged Vph1 (third

*Figure 5 continued on next page*

*Figure 5 continued*

row from top) are shown in mock treated and IAA treated cells. Representative mid-sections are shown. Scale bar = 5 μM (d) Quantification shows an increase in the number of cells where Dnf2 co-localizes with vacuoles after IAA induced degradation of Vps53 up to 15% after 90 min. Error bars show the standard deviation of 3 different experiments. (e) Fold change Pho8 activity of plasma membrane reporters for sorting into vacuoles. Fold change was calculated as the ratio of drug treated and control samples. (IAA for Dnf2, Lem3, Pdr12 and inositol for Itr1). Error bars represent standard deviations from three to five independent experiments.

DOI: https://doi.org/10.7554/eLife.42837.012

The following figure supplement is available for figure 5:

**Figure supplement 1.** Colocalization studies of Lem3, Dnf2, Vps10 and Pdr12 with vacuoles after IAA induced degradation of Vps53.
DOI: https://doi.org/10.7554/eLife.42837.013

flippases to the vacuole suggests that also the phospholipid homeostasis in GARP deleted cells is disturbed. To test this hypothesis we isolated lipids from IAA treated OsTir Vps53-AID-6HA and OsTir Vps53-6HA cells without an AID tag and analyzed the levels of the most abundant phospho-glycerolipids phosphatidic acid (PA), phosphatidyl-serine (PS), phosphatidyl-inositol (PI), phosphati-dyl-ethanolamine (PE) and phosphatidylcholine (PC) and phosphpatidyl-glycerol (PG) as well as the sphingolipid intermediates long chain bases (LCB) and ceramides (CER) (*Figure 6c*). Interestingly, we observed significant changes in the PC to PE ratio with a significant change in PE levels (p=0.04448). These are the two lipid classes that are flipped across the bilayer by Dnf1 and Dnf2 (*Stevens et al., 2008*). This suggests that the depletion of the GARP complex and the concomitant loss of flippases from the plasma membrane results in changes in the overall phospholipid composition. We also detected a significant 1.5 fold increase in LCBs in GARP depleted cells (p=0.01525). Although similar to GARP knockouts, this is a much smaller increase then we had detected previously for GARP knockout mutants (*Fröhlich et al., 2015*).

To test if we see any changes in the lipid composition of the vacuole we purified vacuoles from IAA treated OsTir Vps53-AID-6HA and OsTir Vps53-6HA cells without an AID tag and analysed their lipid composition by MS based lipidomics. In this case we did not detect any significant changes for LCB levels nor for the general phospholipid composition. The only exception to this was phosphati-dyl-serine which is significantly reduced in vacuoles of GARP depleted cells (p=0.03305; *Figure 6d*).

Next, we tested whether sphingolipid levels affect the trafficking of flippases in GARP depleted cells. Therefore, we chemically depleted sphingolipids by addition of myriocin in control- or IAA treated OsTir VPS53-AID-6HA cells. Depletion of sphingolipid levels by addition of myriocin pre-vented the re-localization of Lem3-mNeon from the plasma membrane to the vacuole (*Figure 6e*).

## Discussion

Here we developed a novel assay based on auxin induced degradation of the GARP complex fol-lowed by MS based analysis of the vacuolar proteome and lipidome. This assay allows identification of proteins targeted to the vacuole instead of being recycled to the plasma membrane in a GARP complex dependent process. We go on to show that two groups of proteins, amino phospholipid flippases and cell wall proteins are specifically mis-targeted to the vacuole. Especially the mis-locali-zation of the several flippases, usually localized to the plasma membrane results in changes in lipid homeostasis.

Mutations in the GARP complex result in a large variety of phenotypes in the yeast, *Saccharomy-ces cerevisiae,* ranging from protein sorting defects (*Conibear and Stevens, 2000*), to defects in autophagy and mitochondrial tubulation (*Conibear and Stevens, 2000*) to sphingolipid homeostasis (*Fröhlich et al., 2015*). The canonical pathway for the GARP complex is the retrograde transport of the CPY receptor Vps10. While it is clear that GARP mutations cause a CPY transport defect it seems unlikely that all the other observed phenotypes are a consequence of this. To understand the com-plexity of the phenotypes of GARP mutations it is necessary to identify the first defects occurring in the cell after GARP depletion.

The assay we developed allows us to observe the changes appearing in the cell as early as 30 min after depletion of the GARP complex. We observe rapidly occurring re-localization of the amino-phospholipid flippases Dnf1 and Dnf2 as well as their adaptor protein Lem3 from the plasma mem-brane to the yeast vacuole. Interestingly, these proteins do not seem to reach the vacuolar lumen,

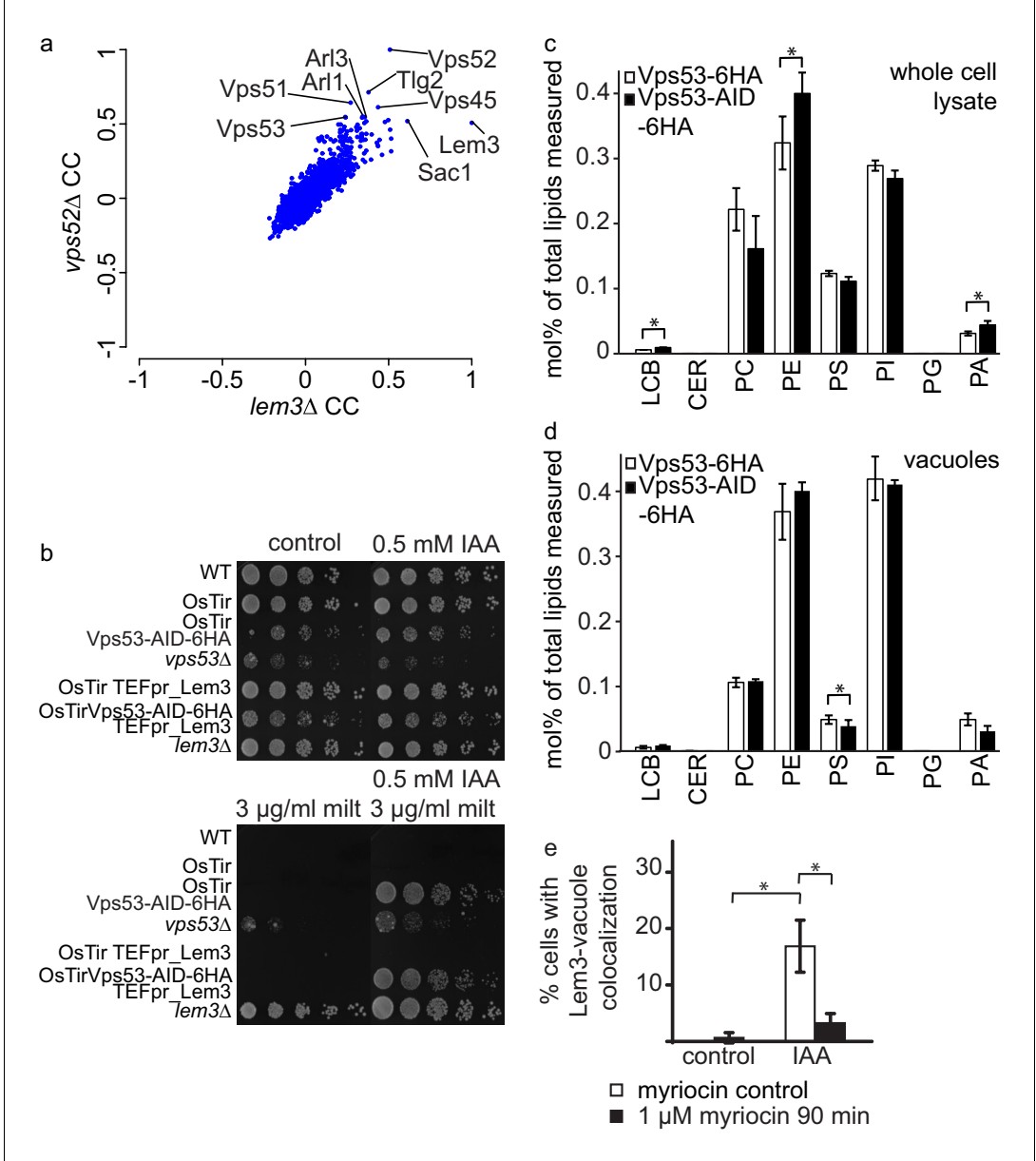

**Figure 6.** GARP depletion pheno-copies the deletion of *LEM3* and affects the cellular lipidome. (a) GARP knockouts and *LEM3* knockouts show highly correlating profiles in chemical genomics datasets. Correlation coefficients (CCs) between the profile of *LEM3* and each other profile in the chemogenetic screen (*Hoepfner et al., 2014*) are plotted on the x-axis. Plotted on the y-axis are the similar sets of values for the *VPS52* profile with all other profiles. (b) Depletion of *VPS53* or *LEM3* results in resistance to the cytotoxic PC analog miltefosine. WT cells, OsTir cells, Vps53-AID OsTir cells, *vps53Δ* cells, OsTir *TEF_LEM3* cells, Vps53-AID OsTir *TEF_LEM3* and *lem3Δ* cells were spotted on control plates (top left panel), plates containing 500 μM IAA (top right panel), plates containing miltefosine (lower left panel) or a combination of IAA and miltefosine (lower right panel). (c) GARP depletion results in changes of the cellular phospholipid composition. The lipidomic analysis of phosphoglycerolipids and sphingolipid intermediates from IAA treated OsTir Vps53-AID-6HA (black bars) or OsTir Vps53−6 HA cells are shown. Long chain bases (LCB), ceramides (CER) phosphatidic acid (PA), phosphatidyl-serine (PS), phosphatidyl-inositol (PI), phosphatidyl-ethanolamine (PE) phosphatidylcholine (PC) and phosphatidylglycerol (PG). (d) GARP depletion has a minor effect on the vacuolar lipidome. Same as c) except that lipids were extracted from enriched vacuoles. Error bars represent standard deviations from three different experiments. (e) Co-localization of mNeon-tagged Lem3 with mCherry-tagged Vph1 in a strain harboring OsTir Vps53-AID-6HA are shown in mock treated (control) and IAA treated cells (IAA). Quantification of 3 different experiments shows co-localization of vacuoles and Lem3 in approximately 15–20% of cells after 90 min IAA treatment compared to control cells (white bars). Additional addition of myriocin prevents Lem3 localization to the vacuole in IAA treated compared to control cells (black bars) (error bars show standard deviations).
DOI: https://doi.org/10.7554/eLife.42837.014

suggesting that the transport carriers do not fuse with the vacuole and are not destined to arrive at the vacuole in WT cells.

Flippases are crucial to maintain phospholipid asymmetry across the plasma membrane (*Hachiro et al., 2013*; *Nakano et al., 2008*). The loss of these proteins in the plasma membrane most likely results in changes of the plasma membrane lipid composition. This note is supported by the changes in the PE to PC ratio we observe in cells. This could also explain why subunits of the GARP complex have been identified in screens using plasma membrane organization as a readout (*Fröhlich et al., 2009*; *Grossmann et al., 2008*). It is also possible that changes in the plasma membrane composition affect the lipid composition of endocytic vesicles derived from the plasma membrane and thus endocytic sorting. Such defects have previously been shown in yeast strains harbouring mutations of flippase proteins (*Hachiro et al., 2013*; *Hua et al., 2002*).

We have previously suggested that the accumulation of sphingolipids causes the vacuolar fragmentation in *vps53Δ* cells. This theory is supported by the fact that depletion of sphingolipids reverses the vacuolar fragmentation defect observed in cells (*Fröhlich et al., 2015*). A recent report also suggests that changes in phospholipids that are transported to the vacuole could result in vacuolar fragmentation (*Ma et al., 2018*). However, we could not detect changes in the vacuolar lipid composition in GARP depleted cells but still observe vacuolar fragmentation as early as 60 mins after depletion of Vps53. It remains possible that our vacuole purification protocol systematically excludes highly fragmented vacuoles where certain lipids are enriched. Our microscopic analysis of purified vacuoles from GARP depleted cells allows this conclusion. It remains also possible that the fragmentation of vacuoles is not the consequence of the accumulation of LCBs but rather depends on different factors. A possible explanation can be a change in ion homeostasis due to changes in plasma membrane composition and thus permeability (*Mioka et al., 2018*). Changes in vacuolar morphology could lead to changes in lipid export, especially LCBs, leading to the massive accumulation we observed previously in GARP knockout cells.

Depleting cellular sphingolipid levels by addition of myriocin results in a block of the re-localization of Lem3 from the plasma membrane to vacuoles. Depletion of sphingolipids after myriocin addition can be detected as early as 10–30 min (*Berchtold et al., 2012*) and is known to block endocytosis in both yeast and mammalian cells (*Pepperl et al., 2013*; *Rispal et al., 2015*; *Zanolari et al., 2000*). Thus, sphingolipid depletion can have two beneficial effects in GARP depleted cells, the inhibition of endocytosis and therefore preventing mis-localization of amino phospholipid flippases. And second, a reduced flux of sphingolipids that reach the vacuole. However, these effects are difficult to dissect biochemically and the method presented here can only detect short term effects. Better protocols for the purification and lipidomic analysis of vacuoles will be crucial to answer these questions in the future. Understanding the lipid related phenotypes will be important to understand the effects of mutations in GARP subunits in human disease (*Feinstein et al., 2014*; *Gershlick et al., 2018*).

Our study also gives new insights in the general organization of the endosomal pathway in yeast. A recent report suggests that yeast has a minimal endosomal system where the trans Golgi network can function as an early/sorting endosome (*Day et al., 2018*). Our data suggest that two classes of proteins, the amino phospholipid flippases and cell wall proteins such as Chs1 are constantly shuttling between the plasma membrane and the Golgi complex. The GARP complex, in this scenario can also be a tether for vesicles directly arriving from the plasma membrane and not only for retrograde endosomal transport carriers. Similar results and GARP dependency have previously been observed for the yeast monocarboxylate transporter Jen1 depending on carbon source availability (*Becuwe and Léon, 2014*). For cell wall maintenance proteins we have evidence that the yeast AP-2 complex is the endocytic adaptor. So far, the function of AP-2 in yeast remains enigmatic. One report suggests the cell wall integrity sensor Mid2 is a cargo of the AP-2 complex and that cell wall maintenance in general is depending on AP-2 (*Chapa-y-Lazo et al., 2014*). While we do not find Mid2 at the vacuole in GARP depleted cells we observe several other cell wall related proteins including Chs1, Cwp1 and Flc2. All these proteins are not accumulating at the vacuole of GARP depleted cells when the AP-2 adaptor complex is knocked out. In contrast, flippase recycling is reported to be dependent on the endocytic adaptor Sla1 (*Liu et al., 2007*). While we were unable to purify vacuoles from *sla1Δ* cells, this suggests that several endocytic pathways are dependent on tethering by the GARP complex. In mammalian cells transport of the lysosomal Niemann Pick Type C protein 2 (NPC2) depends on a functional GARP complex and mannose-6-phosphate receptor

recycling (*Wei et al., 2017*). We have not observed any trafficking defects for yeast Npc2 in our experiments. It was previously shown that the localization of yeast Npc2 depends on a functional AP-3 pathway (*Berger et al., 2007*) and thus it remains possible that we cannot observe defects depending on a functional GARP recycling pathway. Another explanation is the long half-life of 7.1 hr measured experimentally for Npc2 (*Christiano et al., 2014*). Changes in the abundance of a vacuole resident protein with a slow turnover rate will probably not be detected in a time course of GARP depletion over 90 min. However, it remains possible that the GARP complex also plays a role in Npc2 transport to the yeast vacuole but in this study we focus on the events occurring early after GARP depletion.

Finally, we anticipate our assay to be a very useful tool to systematically study the endosomal sorting pathway. The combination of vacuole purification and mass spectrometry proteomics can be used to determine cargo of several sorting complex. For example, acute depletion of the AP3 complex (*Llinares et al., 2015*) should result in the lack of proteins delivered to the vacuole via this pathway. Alternatively, the assay could be adapted to identify cargo of the AP-2 complex that is localized to the yeast plasma membrane (*Rad et al., 1995*). But in contrast to its mammalian counterpart its function and cargo remain largely elusive.

## Materials and methods

### Yeast strains and plasmids

Yeast strains used in this study are described in *Supplementary file 1*. Plasmids used in this study are summarized in *Supplementary file 2*.

### Yeast media and growth conditions

Yeast strains were grown according to standard procedures. For spotting assays, myriocin, IAA and miltefosine were added at concentrations as indicated and the plates were incubated at 30°C for 48 hr.

For SILAC labeling procedures yeast cells were grown in SDC-lysine medium consisting of 2% glucose, 6.7 g/L yeast nitrogen base without amino acids and yeast synthetic dropout without lysine (Sigma Aldrich). Pre cultures were grown over night in the presence of 30 mg/L normal lysine or heavy lysine (L-Lysine $^{13}C_6^{15}N_2$; Cambridge Isotope Laboratories) and diluted to $OD_{600}$ = 0.1. Cells were grown to $OD_{600}$ = 0.5–1 before harvest.

### Vacuole isolation

For Western blot analysis vacuoles were purified from 1L YPD culture. Cells were incubated with 500 µM IAA or same amount abs. EtOH (for negative ctrl) at $OD_{600}$ = 0.5–0.8 for 90 min at 30°C. The logarithmic phase cells were harvested with centrifugation and the pellet treated with Tris-buffer (0.1 M Tris, pH 9.4; 10 mM DTT) and spheroblasting buffer (0.6 M sorbitol, 50 mM KPi, pH 7.4, in 0.2x YPD). After lyticase digestion, vacuoles were isolated via dextran lysis and Ficoll gradient flotation (*Cabrera and Ungermann, 2008*). 500 µl of the 0–4% interphase were taken and mixed with 25 µl 20x PIC. Protein concentration was measured with Bradford assay. For MS experiments vacuoles were purified from 500 ml SDC-Lys medium. To compare two settings one strain grown in SDC-Lys +heavy lysine (K8, 30 µg/ml final) and the other in SDC-Lys +light lysine (K0, 30 µg/ml final) at 30°C. Cultures were treated with 500 µM IAA or ethanol (as ctrl) for 30–90 min. Before centrifugation same OD-units of both cell cultures were mixed and harvested together.

### Fluorescence microscopy

Cells were grown to logarithmic phase in synthetic medium, supplemented with essential amino acids (SDC). IAA was added at concentrations indicated. Cells were imaged live in SDC media unless stated otherwise on an Olympus IX-71 inverted microscope equipped with 100x NA 1.49 and 60x NA 1.40 objectives, a sCMOS camera (PCO, Kelheim, Germany), an InsightSSI illumination system, 4′,6-diamidino-2-phenylindole, GFP and mCherry filters, and SoftWoRx software (Applied Precision, Issaquah, WA). We used constrained-iterative deconvolution (SoftWoRx). All microscopy image processing and quantification was performed using ImageJ (National Institutes of Health, Bethesda, MD; RRID:SCR_003070).

## Western blot

For Western blot comparison of vacuole and cell samples purified vacuoles or whole cell lysate was used. For cell lysate samples 250 µl RIPA buffer (25 mM Tris-HCl pH 7.6, 150 mM NaCl, 1% NP-40, 1% Na-Deoxycholate, 0.1% SDS) and 500 µl zirconia beads were added to the cell pellet in a 1.5 ml reaction tube and lysed for 40 s at 4°C with the Fast Prep system (MP biomedicals). The tubes were pierced at the bottom and the lysate centrifuged at 4000 rpm and 4°C for 30 s in a new tube. Lysate was centrifuged again for 5 min at 14.000 rpm and the supernatant used for Western blot. Protein concentration was determined via Bradford assay. Samples were analyzed by western blotting. HA tagged proteins were detected with a mouse anti-HA antibody 12CA5 (Roche; RRID:AB_514505) diluted 1:2000, Pgk1 (Thermo; RRID:AB_2532235) using a 1:20000 dilution of a mouse antibody and horseradish peroxidase coupled mouse IgG kappa binding protein (Santa Cruz biotechnology; RRID: AB_2687626).

## Proteomics

Mass spectrometry was done with purified vacuoles and whole cell lysate, vacuoles were further purified by in-gel digest and cell lysate samples by Filter Aided Sample Preparation (FASP; *Wiśniewski et al., 2009*). Purified vacuole samples were precipitated with 100% TCA and the protein pellet washed with acetone. The pellet was solved in 4x loading dye and loaded on a 10% denaturating SDS-gel for some minutes. All following steps were performed in glass vials. Gel pieces with proteins were cut and incubated in destaining buffer (25 mM $NH_4HCO_3$(ABC)/50% EtOH) twice for 20 min at 25°C under shaking. After Dehydration in 100% EtOH (twice for 10 min at 25°C) and drying the gel pieces were rehydrated in reduction buffer (10 mM DTT in 50 mM ABC) for 60 min at 56°C followed by alkylation (55 mM iodoacetamide in 50 mM ABC) for 45 min at 25°C in the dark and another washing step for 20 min with digestion buffer. After dehydration in EtOH (10 min, 25°C) and washing with digestion buffer (50 mM $NH_4HCO_3$ in water, pH 8.0, 20 min, 25°C) gel pieces were again incubated twice with EtOH for 10 min and dried. Gel pieces were rehydrated in LysC solution (final 16 µg/ml in 50 mM ABC) for 20 min at 4°C, the excess of solution was removed, digestion buffer added and the sample incubated over night at 37°C. Digestion was stopped by adding 2 µl 100% TFA. Gel pieces were incubated twice in extraction buffer (3% TFA/30% ACN) for 10 min at 25°C and twice with ACN for 10 min at 25°C. The supernatants were collected and dried until most of the solvent was gone and resolved in 50 µl HPLC-grade water. Cell lysate pellets were lysed in 200 µl lysis buffer (Tris 0.1 M, pH 9; 0.1 M DTT; 5% SDS) for 30 min at 55°C and mixed with 1.2 ml 8 M urea in 0.1 M Tris/HCl pH 8.5 (UA). The cell lysate was centrifuged in a wet filter unit (30.000K) for 15 min at 14.000 rpm and the filter washed four times with 200 µl UA for each 10 min. 200 µl IAA solution (0.05 M iodoacetamide in UA) was added to the filter units, shaken vigorously for 1 min and then incubated for 20 min without mixing in the dark. Samples were washed four times with UA for 10 min at 14.000 rpm and washed again with 50 mM ABC and three times with 20 mM. Reversed-phase chromatography was performed on a Thermo Ultimate 3000 RSLCnano system connected to a Q Exactive*Plus* mass spectrometer (Thermo) through a nano-electrospray ion source. Peptides were separated on 50 cm PepMap C18 easy spray columns (Thermo) with an inner diameter of 75 µm. The column temperature was kept at 40°C. Peptides were eluted from the column with a linear gradient of acetonitrile from 10–35% in 0.1% formic acid for 118 min at a constant flow rate of 300 nl/min. Eluted peptides from the column were directly electrosprayed into the mass spectrometer. Mass spectra were acquired on the Q Exactive*Plus* in a data-dependent mode to automatically switch between full scan MS and up to ten data-dependent MS/MS scans. The maximum injection time for full scans was 50 ms, with a target value of 3,000,000 at a resolution of 70,000 at m/z = 200. The ten most intense multiply charged ions (z = 2) from the survey scan were selected with an isolation width of 1.6 Th and fragment with higher energy collision dissociation (*Olsen et al., 2007*) with normalized collision energies of 27. Target values for MS/MS were set at 100,000 with a maximum injection time of 80 ms at a resolution of 17,500 at m/z = 200. To avoid repetitive sequencing, the dynamic exclusion of sequenced peptides was set at 30 s. The resulting MS and MS/MS spectra were analyzed using MaxQuant (version 1.6.0.13, www. maxquant.org/; [*Cox and Mann, 2008*; *Cox et al., 2011*] as described previously [*Fröhlich et al., 2013*]). All calculations and plots were performed with the R software package (www.r-project.org/; RRID:SCR_001905)

## Lipidomics

For the LC-MS/MS analysis, lipids were extracted from lysed yeast cells or purified vacuoles according to 12 µg of protein by chloroform/methanol extraction (*Ejsing et al., 2009*). Prior to extraction a standard mix containing phosphatidic acid (PA 17:0/14:1), phosphatidylserine (PS 17:0/14:1), phosphatidylinositol (PI 17:0/14:1), phosphatidylethanolamine (PE 17:0/14:1), phosphatidylglycerol (PG 17:0/14:1), phosphatidylcholine (PC 17:0/14:1); sphingosine (LCB 17:0) and ceramide (CER 18:0/17:1)was spiked into each sample for normalization and quantification. Dried lipid samples were dissolved in a 65:35 mixture of mobile phase A (60:40 water/acetonitrile, including 10 mM ammonium formate and 0.1% formic acid) and mobile phase B (88:10:2 2-propanol/acetonitrile/$H_2O$, including 2 mM ammonium formate and 0.02% formic acid). HPLC analysis was performed employing a C30 reverse-phase column (Thermo Acclaim C30, 2.1 $\times$ 250 mm, 3 µm, operated at 50° C; Thermo Fisher Scientific) connected to an HP 1100 series HPLC (Agilent) HPLC system and a QExactive*PLUS* orbitrap mass spectrometer (Thermo Fisher Scientific) equipped with a heated electrospray ionization (HESI) probe. The elution was performed with a gradient of 45 min; during 0–3 min, elution starts with 40% B and increases to 100%; in a linear gradient over 23 mins. 100% B is maintained for 3 mins. Afterwards solvent B was decreased to 40% and maintained for another 15 min for column re-equilibration. The flow-rate was set to 0.1 ml/min. MS spectra of lipids were acquired in full-scan/data-dependent MS2 mode. The maximum injection time for full scans was 100 ms, with a target value of 3,000,000 at a resolution of 70,000 at m/z 200 and a mass range of 200–2000 m/z in both, positive and negative mode. The 10 most intense ions from the survey scan were selected and fragmented with HCD with a normalized collision energy of 30. Target values for MS/MS were set at 100,000 with a maximum injection time of 50 ms at a resolution of 17,500 at m/z 200. To avoid repetitive sequencing, the dynamic exclusion of sequenced lipids was set at 10 s. Peaks were analyzed using the Lipid Search algorithm (MKI, Tokyo, Japan). Peaks were defined through raw files, product ion and precursor ion accurate masses. Candidate molecular species were identified by database (>1,000,000 entries) search of positive (+$H^+$; +$NH_4^+$) or negative ion adducts (-$H^-$;+$COOH^-$). Mass tolerance was set to five ppm for the precursor mass. Samples were aligned within a time window and results combined in a single report. From the intensities of lipid standards and lipid classes absolute values for each lipid in pmol/mg protein were calculated. Data are displayed as mol% of total lipids measured.

## Pho8-Assay

Cells are grown to log phase in SDC medium (or SDC-inositol for control samples) and were incubated with 500 µM IAA or EtOH for 90 min (or 1 mM inositol for 30 min, control samples). A cell pellet equivalent to 3 OD units was washed with water and again with ice cold 0.85% NaCl containing PMSF. The supernatant was removed and the pellet resuspended in 300 µl ice cold lysis buffer (20 mM PIPES, 0.5% Triton X-100, 50 mM KCl, 100 mM potassium acetate, 10 mM $MgSO_4$, 10 µM $ZnSO_4$, 1 mM PMSF). The cells were lysed with glass beads in a Fast Prep Homogenizer at 4°C. After centrifugation 100 µl of the supernatant were mixed with 400 µl prewarmed reaction buffer (125 mM p-nitrophenyl phosphate (pNPP), 250 mM Tris-HCl, pH 8.5, 0.4% Triton X-100, 10 mM $MgSO_4$, 10 µM $ZnSO_4$). Blanks were performed with 100 µl of reaction buffer instead of cell lysate. Samples were incubated at 37°C for 15–25 min and the reaction stopped with 500 µl stop solution (1M glycine/KOH, pH 11.0).The samples were centrifuged at maximal speed for 2 min and the absorbance measured at 400 nm (*Klionsky, 2007*).

## Acknowledgements

We thank members of the Fröhlich lab for discussions and careful reading of the manuscript. We thank Robbie Loewith, David Teis and Christian Ungermann for sharing of reagents. We thank Stefan Walter for support and maintenance of the mass spectrometer. Florian Fröhlich is supported by the DFG grant FR 3647/2–1 and the SFB944.

## Additional information

### Funding

| Funder | Grant reference number | Author |
| --- | --- | --- |
| Deutsche Forschungsgemeinschaft | FR 3647/2-1 | Florian Fröhlich |

The funders had no role in study design, data collection and interpretation, or the decision to submit the work for publication.

### Author contributions

Sebastian Eising, Data curation, Formal analysis, Investigation; Lisa Thiele, Data curation, Formal analysis, Investigation, Methodology; Florian Fröhlich, Conceptualization, Resources, Data curation, Software, Formal analysis, Supervision, Funding acquisition, Investigation, Methodology, Writing—original draft, Project administration, Writing—review and editing

### Author ORCIDs

Florian Fröhlich (iD) http://orcid.org/0000-0001-8307-2189

### Decision letter and Author response

Decision letter https://doi.org/10.7554/eLife.42837.020
Author response https://doi.org/10.7554/eLife.42837.021

## Additional files

### Supplementary files

• Supplementary file 1. List of all yeast strains used in this study.
DOI: https://doi.org/10.7554/eLife.42837.015

• Supplementary file 2. List of all plasmids used in this study.
DOI: https://doi.org/10.7554/eLife.42837.016

• Transparent reporting form
DOI: https://doi.org/10.7554/eLife.42837.017

### Data availability

All data generated or analysed during this study are included in the manuscript and supporting files. Source data files have been provided for Figures 3 and 4.

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
