## [Decision Letter]

Thank you for submitting your article "A Systematic Approach to Identify Recycling Endocytic Cargo Depending on the GARP Complex" for consideration by *eLife*. Your article has been reviewed by three peer reviewers, one of whom is a member of our Board of Reviewing Editors, and the evaluation has been overseen by a Reviewing Editor and Vivek Malhotra as the Senior Editor. The following individual involved in review of your submission has agreed to reveal her identity: Elizabeth A Miller (Reviewer #2).

The reviewers have discussed the reviews with one another and the Reviewing Editor has drafted this decision to help you prepare a revised submission.

Summary:

This research advance builds on a previous *eLife* paper investigating the cellular function of the GARP complex. The authors created an acute GARP depletion model via auxin-induced degradation and examined the vacuolar proteome to determine which proteins use the GARP pathway. The consensus opinion was that results are convincing and that the technique described here will aid the field in dissecting the details of protein secretion and traffic.

Essential revisions:

1) Regarding the AP-2 dependence effects, is it possible that Dnf1/Dnf2/Lem3 is unaffected by *apl1* deletion because of a population that is not at the plasma membrane but instead is diverted from an intracellular pool?

2) Regarding cause/effect of lipid alterations: are the observed trafficking defects of Dnf1/Dnf2/Lem3 causative of sphingolipid effects in GARP KO cells, or are they effects? One might dissect this by using chemical depletion of sphingolipids as the authors did previously in the GARP KO cells. Dnf1/Dnf2 trafficking changes should be unaffected by myriocin if they are the direct targets of GARP. The reviewers consider this an important control.

3) The results here seem to differ from findings in mammals (see https://www.ncbi.nlm.nih.gov/pubmed/28658628). If feasible, it would be ideal if the authors could validate the results by knocking down the GARP complex in a mammalian cell; but at the very least this apparent discrepancy should be discussed in depth.

---

## [Author Response]

Summary:This research advance builds on a previous eLife paper investigating the cellular function of the GARP complex. The authors created an acute GARP depletion model via auxin-induced degradation and examined the vacuolar proteome to determine which proteins use the GARP pathway. The consensus opinion was that results are convincing and that the technique described here will aid the field in dissecting the details of protein secretion and traffic.Essential revisions:1) Regarding the AP-2 dependence effects, is it possible that Dnf1/Dnf2/Lem3 is unaffected by apl1 deletion because of a population that is not at the plasma membrane but instead is diverted from an intracellular pool?

We would like to thank the referees for raising this important point. We agree that our mass spectrometric data do not rule out this possibility. We have now confirmed our results by live cell imaging and added an analysis of the intracellular distribution of the Lem3 signal. While this still does not rule out divertion from an intracellular pool, it suggests that the plasma membrane pool of Lem3 gets also depleted. We have added these data as Figure 4—figure supplement 5 and have modified the text accordingly.

2) Regarding cause/effect of lipid alterations: are the observed trafficking defects of Dnf1/Dnf2/Lem3 causative of sphingolipid effects in GARP KO cells, or are they effects? One might dissect this by using chemical depletion of sphingolipids as the authors did previously in the GARP KO cells. Dnf1/Dnf2 trafficking changes should be unaffected by myriocin if they are the direct targets of GARP. The reviewers consider this an important control.

The referees raise an interesting point. We have carried out the suggested experiments to test Lem3 localization upon GARP depletion with and without sphingolipid biosynthesis inhibition. Interestingly, Lem3 remains plasma membrane localized in GARP depleted cells upon myriocin treatment. However, in our view, this does not rule out that flippases/Lem3 as direct targets of the GARP complex. Chemical sphingolipid depletion is known to lead to actin deploimerization and inhibition of endocytosis. Therefore, the plasma membrane localized proteins remain in the plasma membrane even when the GARP complex is depleted. We have previously shown that deletion of *VPS53* results in an accumulation of long chain bases and chemical sphingolipid biosynthesis is beneficial for the cells. In the auxin induced degron system that we use here, we can only detect small changes in sphingolipid levels. However, the beneficial effects of myriocin treatment in VPS53 knockout/depleted cells could be both(i) suppression of endocytosis and thus flippase activity in the plasma membrane (ii) lower levels of sphingolipids reaching the yeast vacuole. We have added the new data to Figure 6 and modified the Results section and Discussion section accordingly.

3) The results here seem to differ from findings in mammals (see https://www.ncbi.nlm.nih.gov/pubmed/28658628). If feasible, it would be ideal if the authors could validate the results by knocking down the GARP complex in a mammalian cell; but at the very least this apparent discrepancy should be discussed in depth.

We agree with the referees that this important discrepany should be addressed in the manuscript. We therefore expressed V5 tagged CDC50A and CDC50B as well as HARFP tagged ATP8B1 in HeLa cells from plasmids and depleted VPS53 by siRNA. We could only detect CDC50A and CDC50B as well as ATP8B1 signal in the ER which is probably an artifact of the massive overexpression (Figure for the referees). However, NPC2 is described to be transported to the yeast vacuole in an AP3 dependent pathway. Additionally, the long half-life (7.1 hours) of Npc2 makes it difficult to detect changes in protein abundance after 90 minutes of GARP depletion. We have edited our Discussion section to include these very important points.